

# Event based high resolution measurement of DOC-concentration and quality in a rural headwater catchment.

Lukas S. Ditzel[1], Caroline Spill[1], Matthias Gassmann[1]

[1]Department of Hydrology and Substance balances, University Kassel, Kassel, Germany

*Correspondence to*: Lukas S. Ditzel (lukas.ditzel@uni-kassel.de)

**Abstract.** The export of dissolved organic carbon (DOC) from small river systems is a significant factor in the global carbon cycle. DOC quality can be used to identify the sources of carbon in headwater systems. High-resolution in-situ measurements in small headwater catchments can unveil fast changing patterns of DOC export and DOC quality during events. In this study, the influence of precipitation events on DOC export and changing

DOC quality was analyzed using highly resolved discharge and DOC concentration and quality data of 5-minute time steps. Data analysis was conducted using spearman correlation analysis, hysteresis analysis and visual comparison of pre-event and event conditions. Measurements took place from January 2021 until August 2022 in a German lower mountain catchment with predominantly agricultural land use. While DOC export was lower than in other catchments the DOC quality followed a well-observed seasonal pattern and was significantly

influenced by the antecedent wetness of the catchment and the length of precipitation events. The results showed that the use of in-situ high resolution measurements can provide a detailed insight into the DOC export dynamics of a catchment and can help to identify the most important drivers of DOC quality changes.

## 1 Introduction

Dissolved organic carbon (DOC) is a significant part of dissolved organic matter (DOM). As a basic nutrient it

contributes to the carbon budget of riverine systems but also as a basic nutrient (Hansen et al., 2016; Gabor et al., 2015; Cole et al., 2007). The role of riverine DOC and the corresponding carbon export by bigger rivers is discussed widely in recent studies (Marín-Spiotta et al., 2014; Hossler and Bauer, 2012; Holmes, R, M. et al., 2012). Research suggests, that good quality DOC, characterized by resilient and highly aromatic DOC-molecules, can lead to problems within wastewater treatment plants like increased sludge production (Ritson et

al., 2017; Broder et al., 2017; Sharp et al., 2006). However, molecules with too low quality can lead to similar problems (Herzsprung et al., 2017). Additionally, low quality DOC, in which the aromatic rings are broken, displays a high reactivity and forms bonds with potential phytotoxic heavy-metals such as copper (Guggenberger et al., 1994; Kalbitz et al., 2000; Ren et al., 2015). Low quality DOC in riverine systems is a result of



degradation processes of good quality DOC. These processes include photobleaching (Helms et al., 2008; Moran et al., 2000), microbial activity and soil adsorption processes (Gabor et al., 2015). A common metric to quantify the degradation of DOC, called specific ultraviolet absorbance at 254 nm (SUVA254), was developed by Weishaar et al. (Weishaar et al., 2003). The SUVA254 index quantifies the strength of the aromatic bounds found in a sample of DOC (Hansen et al., 2016). Based on this approach, additional indices quantifying aromaticity were developed, such as the specific visible absorbance at 280 nm (Chin et al., 1994). Whereas spectral slope (SLx) indices like $SL_{275-295}$ and $SL_{290-350}$ quantify the molecular weight of DOC (Helms et al., 2008; Blough and Vecchio, 2017), spectral slope ratios (Helms et al., 2008) additionally give an indication of DOC exposure to irradiation. While many studies on DOC quality have been performed on higher-order streams (Williamson et al., 2021) or forested headwater catchments (Butturini et al., 2006; Werner et al., 2019), there is a scarcity in studies performed on headwater catchments with mainly agricultural land use. In addition, the availability of high frequency data, which is essential for identifying the fast-changing conditions within headwater rivers, is very rare, as there are only a few studies, which installed such measurements in headwater catchments (Spill et al., 2023).

In temperate climates, the process of DOC degradation is depending on the land use (Graeber et al., 2012; Stedmon et al., 2006) and varies with the seasons due to changing soil and water temperatures and solar radiation (Christ and David, 1996; Jaffé et al., 2008). Generally, the production of DOC in temperate climates is higher in the summer months (Hinton et al., 1997; Laudon et al., 2004), when microbial and biomass production rises, and lower in the winter months, when biological activity is slowed down (Hinton et al., 1997). However, the DOC produced in the winter months is estimated to have a better quality and a longer resilience of its aromatic rings (Ritson et al., 2017). Although the DOC-quality tends to be better in winter, the higher soil and water temperatures during the summer months are responsible for generally higher DOC production (Hinton et al., 1997; Winterdahl et al., 2016). But since this freshly produced DOC is more exposed to the higher solar radiation in the summer months, its degradation processes are accelerated. While photobleaching won't occur in groundwater, it is a significant driving factor for degradation in surface waters during the warmer months (Osburn et al., 2009). Small headwater catchments normally don't consist of big surface water areas and are mainly supplied by their local groundwater sources and precipitation. Therefore, the change of DOC quality in such hydrological systems is mainly driven by the seasonal change of the DOC input (Broder et al., 2017; Burd et al., 2018). This DOC input strongly correlates with the land use of the catchment (Graeber et al., 2012). Different types of land use produce different amounts of the compounds of DOC and therefore change the composition of DOC (Hood et al., 2006; Yamashita et al., 2011; Graeber et al., 2012; Lu et al., 2014; Williams et al., 2016; Ritson et al., 2019; Pisani et al., 2020). Additionally, the land use in a temperate climate catchment





typically not only reacts to the varying temperature over the seasons, but the land use itself can often drastically change (Hood et al., 2006). This is because many catchments are in areas heavily influenced by humans and a mainly agricultural catchment can be a forested catchment in a few decades (Williams et al., 2016). On a smaller timescale, land use change can occur due to changing crop rotations or shifting pasture to farmland and vice

versa.

The precipitation input signal can also change the composition of DOC in the river. Precipitation events can mobilize fractions of DOC in the soil and on the soil surface and flush them into the stream (Hinton et al., 1997; Broder et al., 2017; Buffam et al., 2001). These fractions are changing the usual DOC composition of the river for a short time (depending on the catchment), resulting in unexpected DOC concentrations and quality signals.

Because DOC quality changes with land use, precipitation, soil type and season, it is very unlikely to measure the exact same quality between different catchments (Graeber et al., 2012). Even catchments with very similar land use can show significant differences in the DOC quality signal due to different precipitation regimes or mean temperatures (DOC production) or solar radiation (photobleaching) (Helms et al., 2013).

This study evaluates the findings of a measurement campaign, which used a high-resolution (5 min resolution)

in-situ approach for the detection of DOC in the discharge. Approaches using this kind of high-resolution in-situ techniques in low order catchments are quite novel and have been conducted in a similar matter by Werner et. al. (2019) and Ritson et.al (2022) showed the advantage of such sensors, when measuring DOC loads, over grab samples. The campaign took place in a small agricultural headwater catchment.

The aims of this study are to:

1)  Detect and visualize changing patterns in the DOC-quality signal during precipitation events.
2)  Quantify the carbon export and evaluate the carbon export dynamics during the most relevant precipitation events.
3)  Identify the most relevant drivers for changes in the DOC-quality and DOC-export signal during

85        precipitation events by performing correlation analysis.

## 2 Material and Methods

### 2.1 Study site

The Nesselbach catchment (51° 26' 54'' N, 9° 22' 59'' E) was already described by Spill et al. (2023). It is located in the low mountain range of the 'westhessische Senke' which is part of the central European uplands.



The Nesselbach stream contributes to the river Esse which is a tributary of the Diemel river and finally discharges into the North Sea via the Weser River.

The monitored area (3.2 km²) includes the largest part of the Nesselbach catchment, which covers 5.2 km² and drains continuously with a mean discharge of 8.1 l/s throughout the year. The length of the river till the monitoring station is 1.8 km. The elevation of the sub-catchment ranges from 173 m to 306 m, the bedrock

mainly consists of Muschelkalk and Buntsandstein. Soils were classified as mainly Vertisol (90%) with some additional brown earth and rendzina in the forested regions. With 65%, agriculture is dominant in the area, followed by forest (25%) and settlement/commerce (10%). Further details regarding water quality and quantity can be found in Spill et al., (2023).

Meteorological data was collected from August 2020 to November 2022 near the catchment area at 51° 28 ' 2'' 

N, 9° 22' 41'' E, using a WMO-conform weather station (Thies Clima, DL-15). Mean annual precipitation in the monitored time frame was 635mm/a. The mean temperature was 8.6°C. The region has a temperate climate. There were some gaps in the climate data due to temporary energy shortages in the area, these gaps were filled with the nearest DWD (German weather service) weather stations using inverse distance interpolation.



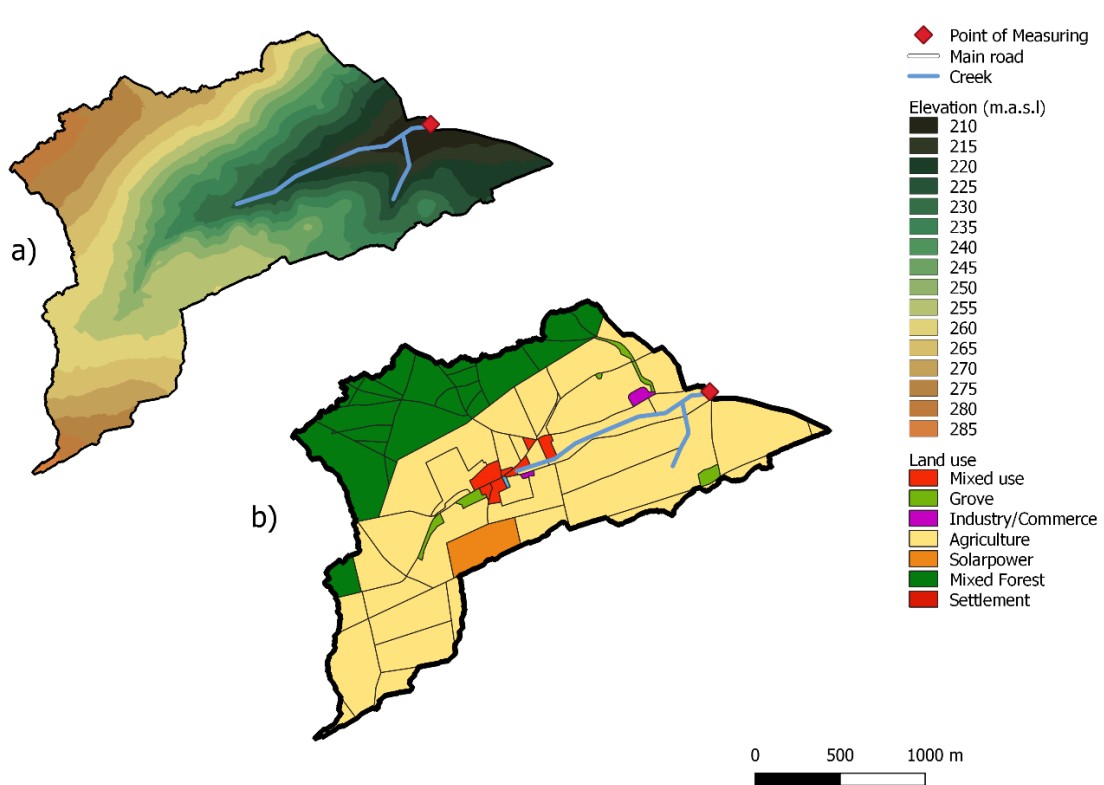

**Figure 1 Study area. a) digital elevation model (elevation in m.as.l.), raster cell resolution 1m. b) Land-use in the study area, resolution 1:10000**

**2.2 In-situ measurement**

Sampling of the DOC concentration and quality was conducted in-situ via a Scan::Spectrolyser UV-VIS (S::can) probe with 5 minutes timesteps. While DOC-concentration was approximated by the device, the quality parameters $SUVA_{254}$ and $SL_{275}$ were delineated from the raw data of the absorption spectra. $SUVA_{254}$ is calculated by equation 1), following Weishaar (2003) and Hansen (2016).

$$SUVA_{254}(\text{L m}^{-1}\text{mgC}^{-1}) = \frac{UV_{254}(\text{cm}^{-1})}{DOC(\text{mgL}^{-1})} \qquad (1)$$

where the absorption peak at 254nm ($UV_{254}$) is divided by the measured DOC-concentration at a given timestep. The calculated $SUVA_{254}$ can be interpreted as an indicator for aromatic strength of DOC molecules (Weishaar et al., 2003).



The indices of spectral slopes for the range from 275nm to 295nm and from 290nm to 350nm were determined by calculating the slope of a linear model fitted to the log-transformed absorption coefficients of the respective ranges of the spectra (Helms et al., 2008). The slopes $SL_{275-295}$ and $SL_{290-350}$ were calculated as a proxy for the molecular weight of DOC (Twardowski et al., 2004). Steeper slope values indicate an increase in degradation

and therefore a weaker aromatic bond and less molecular weight (Helms et al., 2008) and can be used to distinguish between allochthonous and autochthonous DOC (Hansen et al., 2016), if molecular weight is considered to be dependent on microbial degradation and photobleaching. Electrical conductivity (EC) and discharge signals were also sampled every 5 minutes. EC was measured using a Hobo-probe (HOBO U24) calibration data can be found in Spill et.al. (2023). discharge was measured by a Nivus ultrasonic probe, which

delineated total discharge from observed water level and water velocity. Water velocity was approximated internally by using cross-correlated particle density detection. Additionally, discharge was calibrated on multiple occasions using NaCl tracer test. For calibration of the in-situ DOC measurement, grab samples were collected every 2 weeks and an additional ISCO autosampler was installed to sample event-driven discharge with a resolution up to 15 min. Water temperature was also measured in-situ. The complete time series for DOC

concentration was corrected for outliers with a median filter and an additional correction based on abrupt changes of the slope to eliminate measurement errors that are a result of the cleaning of the device or stuck particles in the optical path. Event-scale timeseries were also corrected for outliers, but those were scarce. Outlier correction also took place for the calculated $SUVA_{254}$ and $SL_{275}$ indices. The dataset consists of 149,868 datapoints (DOC, Water temperature) which conforms to roughly 512 sampling days but lacks discharge data on

some occasions.

### 2.3 Discharge data smoothing

The 5 minutes discharge data appeared highly scattered at times. This scattering can be attributed to a lack of particles in the water, which causes uncertainties in the ultra-sonic scanning of particle density. This problem is

visible in the long-term timeseries and in the event driven discharge data, but with less scattering in the latter. The scattering can be tracked throughout the timeseries, but with significantly less scattering from late spring to early autumn. This phenomenon can be explained by an increase in particles in the stream during the warmer months, which is directly linked to biomass growth throughout the summer season. Since the discharge data is needed for correlation-analysis, smoothing of the data was performed. The smoothing was conducted for the 26

events using a Nadayara-Watson kernel-density-estimator (equation 2) with a gaussian kernel (equation 3).



Gaussian kernels are recommended for periodic data and therefore suitable for seasonal data and data with internal periodic inaccuracies (Nadaraya, 1964; Bierens, 1996).

$$\hat{m}(x) = \frac{\sum_{i=1}^{n} y_i K_h(x-x_i)}{\sum_{i=1}^{n} K_h(x-x_i)}$$

(2)

With $\hat{m}(x)$ = estimated regression, $y_i$ = the respond variable of the dataset, $K_h$ = the kernel used for smoothing (equation 4), $x$ = predictor variable and $x_i$ = observation at time step i.

$$K(u) = \frac{1}{\sqrt{2\pi}} \exp\left(-\frac{1}{2}u^2\right)$$

(3)

With $K_h(u) = K\left(\frac{u}{h}\right)/h$ and h>0, where h is the chosen bandwidth and u = $(x-x_i)/h$. To assess the performance

of the smoothing, the Nash-Sutcliffe-efficiency (Nash and Sutcliffe 1970; McCuen et al. 2006) between observed discharge and smoothed discharge was calculated:

$$NSE = 1 - \frac{\sum_{t=1}^{T}(Q_0^t - Q_m^t)^2}{\sum_{t=1}^{T}(Q_0^t - \overline{Q}_0)^2}$$

(4)

With $\overline{Q}_0$ = mean of observed discharges, $Q_m$ = the modeled discharge and $Q_0^t$ = observed discharge at time $t$.

While a NSE of 1 can be interpreted as a complete match between the modeled and the observed discharge, NSE < 0 indicate that the mean of the observed data is a better predictor than the modeled data. To find a suitable ratio between interpretability (and usefulness for later correlation with DOC timeseries) of the discharge function and acceptable information loss, an iterative process was implemented to find the "best" bandwidth for smoothing each individual discharge event. In a first step, the observed discharge data was smoothed with a bandwidth *h*

ranging from 1 to 10 with virtual steps of 0.2. The smoothed observed discharge was Spearman-correlated with the measured height of the water column to test the correspondence of the smoothed data with its origin, assuming that, when water velocity is difficult to measure (due to low particle density), the measuring of the water column was still accurate. The bandwidth with the highest correlation was picked for smoothing the total discharge and NSE was calculated. Only smoothed discharges with NSEs over 0.65 were accepted as suitable

with most of the smoothed timeseries rating significantly above that value.

 After outlier correction, extreme values and arithmetic means were calculated for every parameter in the event-based data set and the whole timeline. The total amount of measured discharge accounts for 385 days.





**2.4 Climate indices**

For additional insights, the event dataset also includes the event length in hours and the wetness conditions in the catchment expressed by the antecedent precipitation index (API) (Fedora and Beschta, 1989). Event length and antecedent precipitation were supposed to have a high influence on DOC export and quality.

$$API_t = (API_{t-\Delta t}K) + P_{\Delta t} \tag{5}$$

With k = recession coefficient, 0.9 for Germany as proposed by Schröter et al. (Schröter et al., 2015), P =
precipitation [mm], and $\Delta t$ = time interval between precipitation observations.

To evaluate the impact of season to the dynamics of DOC export, the discharge-normalized temperature ($DNT_{30}$) (Werner et al. 2019) and the antecedent aridity index ($AI_t$) (Barrow, 1992) were computed. $DNT_{30}$ was calculated to get an estimate of temperature to discharge relations before an event assuming that temperature controls DOC export.

$$DNT_{30} = \frac{\frac{1}{n}\sum_{i=1}^{n=30}(T_i)}{\frac{1}{n}\sum_{i=1}^{n=30}(Q_i)} \tag{6}$$

With n = number of observations, $T_i$ = mean temperature at a given day, $Q_i$ = mean discharge at a given day and $DNT_{30}$ = discharge normalized temperature of the preceding 30 days. The antecedent aridity index was calculated for the preceding 60 days ($AI_{60}$) to get an estimate of the water balance of the catchment (Barrow 1992).

$$AI_{60} = \frac{\sum_{i=1}^{n=60}(P_i)}{\sum_{i=1}^{n=60}(ET_i)} \tag{7}$$

With $P_i$ = precipitation at a given day, $ET_i$ = potential evapotranspiration at a given day, calculated with the Penman-Monteith method. To test for the influences on DOC concentrations and DOC quality, a Spearman-Correlation analysis was conducted on the dataset. The results were controlled for the significance of the paired correlations with a significance level of 0.1.


**2.5 Hysteresis Analysis**

Hysteresis analysis was performed for the discharge-DOC relation, to get deeper insights into the DOC-export dynamics. The rotation of the hysteresis and the hysteresis index HI were calculated following the algorithm of Lawler et al., (2006) They assume that there is a rising and a falling limb of the hydrograph for every event. In a
first step, the mid-point of the event discharge was determined:

$$Q_{mid} = k(Q_{max} - Q_{min}) + Q_{min} \tag{8}$$



With $Q_{mid}$ = mid-point of discharge, $Q_{max}$ = peak discharge, $Q_{min}$ = starting discharge and k = position of representation of loop width. k = 0.5 measures the discharge at the mid-point of the rising limb. After the identification of the midpoint, it is possible to determine the rotation of the hysteresis (Lawler et al., 2006). A clockwise hysteresis is characterized by $DOC_{RL} > DOC_{FL}$ and an anti-clockwise hysteresis by $DOC_{RL} < DOC_{FL}$. Where $DOC_{RL}$ refers to the DOC-concentration on the rising limb and $DOC_{FL}$ refers to the DOC-concentration on the falling limb.

The magnitude of the hysteresis can be quantified by calculating the hysteresis index HI. The sign of the index also indicates the direction of the rotation, where anti-clockwise hysteresis always has negative HI-values. At the mid-point, the hysteresis index $HI_{mid}$ can be calculated as follows depending on the knowledge of the direction of the hysteresis (Lawler et al., 2006):

a) clockwise hysteresis

$$H_{mid} = \left(\frac{TU_{RL}}{TU_{FL}}\right) - 1$$

(9)

b) anti-clockwise hysteresis

$$H_{mid} = (-1/(TU_{RL}/TU_{FL})) + 1 \tag{10}$$

Carbon export mass fluxes were calculated by multiplying the DOC-concentration at a giving time step with discharge, giving:

$$DOC_{export} = Q\left[\frac{l}{s}\right] * DOC\left[\frac{mg}{l}\right] \tag{11}$$

With Q = discharge at a given time step, DOC = DOC concentration at a given time step. The event-based DOC export [kg] was calculated by summing up $DOC_{export}$ values over the event length. Figure 2 shows an example for a typical hysteresis loop observed in this catchment compared to the underlying timeseries of discharge and DOC-concentration.





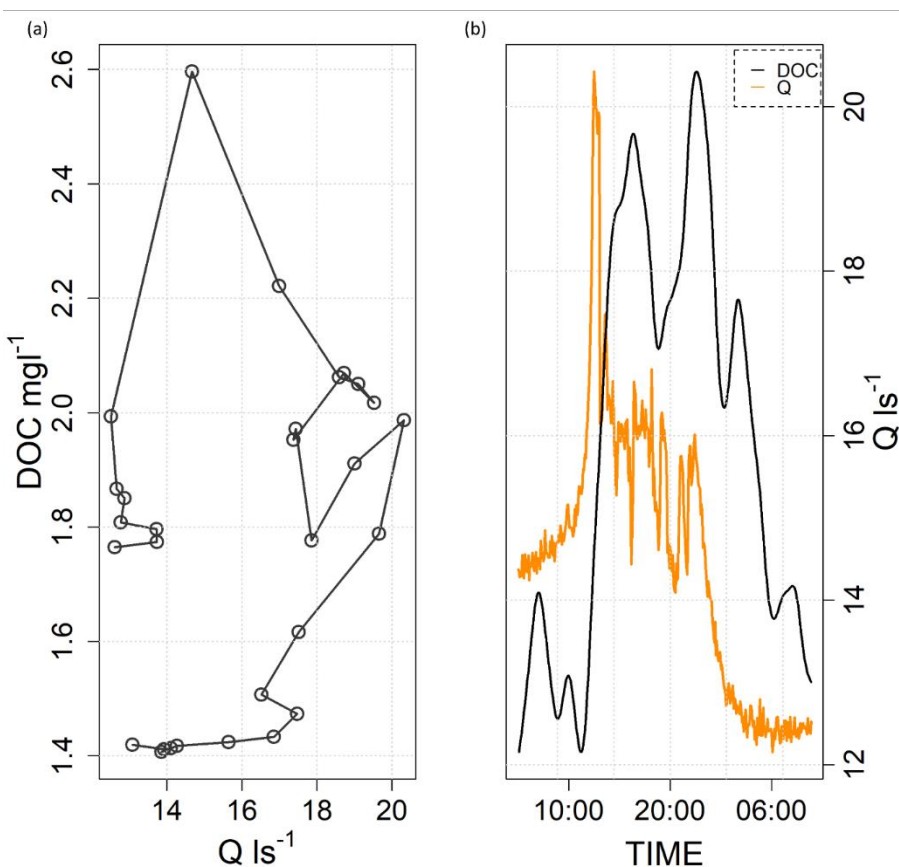

**Figure 2 Example for a clockwise hysteresis loop during a precipitation event on 03 March 2021. The clustering of the start values is quite common for the dataset. Values were aggregated to one hour to gain a better visual overview. b) representation of discharge and nitrate concentration for the same event with a resolution of 5 minutes. Discharge has been smoothed.**

## 3 Results and Discussion

### 3.1 DOC-Quality

Monthly box plots of the SUVA$_{254}$ index, as seen in Fig. 3, show decreasing DOC quality values with the beginning of the spring period: in March, the SUVA254 rapidly decreases and halves its previous values from February to March. After that decline an initially fast and then steady slow recovery, starting in May, can be observed.





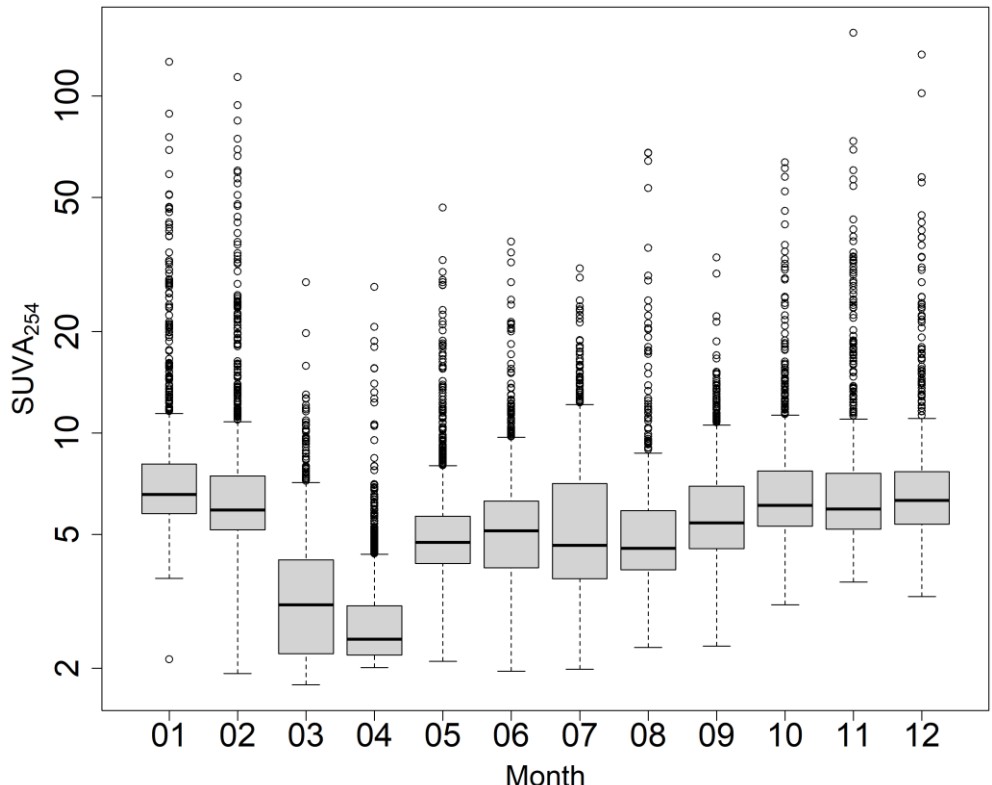

**Figure 3 Daily means of SUVA$_{254}$ [Lm$^{-1}$mgC$^{-1}$], monthly aggregated.**

Figure 4 shows the high-resolution behavior of precipitation, discharge, water temperature, DOC-concentration and SUVA$_{254}$. Each of these parameters exhibits a distinct seasonal pattern over the observed time. Winter

months are characterized by high discharge and precipitation, while summer months see rising temperatures and stronger peaks in DOC and SUVA$_{254}$. This pattern can also be observed in the mean values of each parameter over the duration of precipitation events, as shown in Table 1. Mean discharges during events are generally higher in the winter half year, even though the strongest precipitation event (67.3 mm) was recorded on 14 July 2021 This event also exported the most DOC, had the highest DOC-concentration and the highest DNT$_{30}$ value.

Combined with a mean discharge of 16.2 l*s$^{-1}$ this event has to be rated as an outlier event. During the event with the lowest precipitation (7.3 mm) on 12 October 2021, the highest SUVA$_{254}$ and the lowest SL275 values were measured. The event with the highest discharge and the longest duration was recorded on 02-21-2023. Extreme values of monitored parameters tend to cluster at specific events. This is evident for the events of the14 July 2021, the 12 October 2021and the 31 March 2022where the lowest API occurred together with the lowest

value of SUVA$_{254}$ and the highest spectral slopes. This observed behavior leads to the assumption that the API is



one of the main drivers of DOC-quality. But this could not be confirmed by the correlation analysis since neither the API, nor any of the quality parameters did correlate significantly with each other, as shown in Table 2. Thus, a shift in composition from humic acids to fulvic acids (Helms et al., 2008), attributed to wetter conditions in the catchment was not found. However, API correlates positively with DOC export and event length. The

connection between those parameters is fortified by the strong positive correlation between discharge and DOC export and the positive correlation between discharge and event length. This indicates that events with high DOC export and event length occur during relatively wet conditions in this catchment.

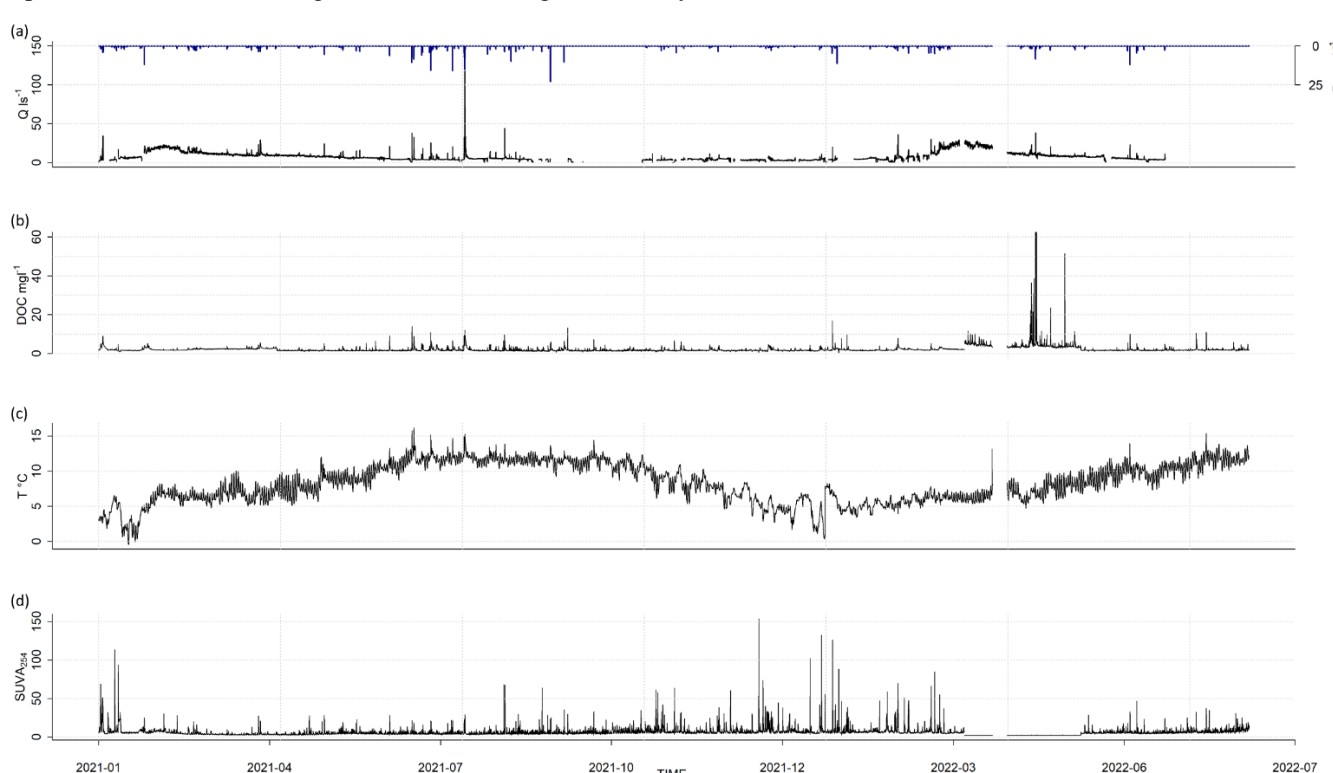

**Figure 4 Aggregated timeseries of a) discharge and precipitation, b) DOC-concentration, c) water temperature, d=**
**SUVA$_{254}$, over the complete measurement campaign. Resolution 1h.**



**Table 1 Mean values of observed parameters during the events picked for further analysis. Extreme values of means are highlighted. Q = event discharge [l/s], T = mean water temperature [°C], DOC = DOC concentration [mg/l], EL = electric conductivity [µS/cm], P = aggregated precipitation [mm], API = API before event [-], SUVA$_{254}$ = specific ultraviolet absorption at 254 nm [ L m$^{-1}$mg-C$^{-1}$], SL$_{275}$ = Spectral slope between 275 and 290 nm [*10$^{-3}$nm$^{-1}$], SL290 = Spectral slope between 290 and 350 nm [*10$^{-3}$nm$^{-1}$], Cexp = Carbon export during event [kg], Length = event length [h].**

| Date | Q | T | DOC | LF | P | API | SUVA$_{254}$ | SL$_{275}$ | SL$_{290}$ | Cexp | Lenght |
|---|---|---|---|---|---|---|---|---|---|---|---|
| 28 January 2021 | 4.296 | 3.367 | 3.098 | 936 | 9 | | 16.687 | 5.809 | **4.813** | 0.015 | 28.583 |
| 08. February 2021 | 6.276 | **1.787** | **1.626** | 968 | 11.8 | | 5.194 | 9.161 | 9.79 | **0.01** | 28.917 |
| 09 March 2021 | 15.982 | 6.091 | 1.785 | | 12.5 | 2.74 | 7.88 | 8.974 | 9.938 | 0.038 | 28.917 |
| 12 March 2021 | 14.159 | 6.32 | 1.692 | 964 | 22.2 | 11.45 | 7.065 | 9.886 | 11.08 | 0.034 | 113.083 |
| 10 April 2021 | 12.116 | 6.793 | 2.252 | 949 | 26 | 8.23 | 6.226 | 10.884 | 11.525 | 0.039 | 84 |
| 10 May 2021 | 10.078 | 9.42 | 2.308 | 911 | 13 | 3.56 | 7.556 | 9.47 | 8.486 | 0.028 | **18.917** |
| 21 June 2021 | 7.38 | **13.161** | 3.401 | 878 | 22.6 | 1.24 | 6.194 | 8.988 | 9.034 | 0.04 | 28.917 |
| 24 June 2021 | 5.197 | 11.922 | 2.649 | 913 | 16.6 | 24.99 | 5.054 | 9.743 | 9.57 | 0.016 | 35.583 |
| 28 June 2021 | 5.657 | 12.481 | 2.978 | 896 | 26.3 | 23.26 | 7.982 | 9.661 | 9.632 | 0.024 | 43.917 |
| 01 July 2021 | 5.511 | 11.622 | 2.661 | 863 | 13.3 | **34.98** | 4.769 | 10.389 | 9.896 | 0.016 | 43.917 |
| 08 July .2021 | 4.622 | 12.019 | 2.286 | 899 | 20.2 | 18.01 | 5.602 | 9.987 | 10.064 | 0.013 | 50.583 |
| 14 July 2021 | 16.292 | 12.932 | **4.347** | 891 | **67.3** | 16.91 | 5.986 | 9.737 | 9.437 | **0.12** | 60.667 |
| 04 August 2021 | 5.453 | 11.53 | 1.974 | **1006** | 13.8 | 7.29 | 10.103 | 8.945 | 9.021 | 0.011 | 40.583 |
| 12 October 2021 | 4.142 | 9.053 | 2.311 | 884 | **7.3** | 2.4 | **17.821** | **5.668** | 5.189 | 0.011 | 18.917 |
| 02 January 2022 | **4.122** | 6.919 | 2.767 | 933 | 10.2 | 4.57 | 12.338 | 6.713 | 6.347 | 0.012 | 40.583 |
| 01 February 2022 | 9.587 | 5.173 | 3.23 | 817 | 16.9 | 9.4 | 10.235 | 7.238 | 5.933 | 0.036 | 49 |
| 06 February 2022 | 8.322 | 5.313 | 2.454 | **776** | 10.8 | 13.96 | 13.301 | 6.24 | 5.09 | 0.021 | 23.917 |
| 17 February 2022 | 16.261 | 6.809 | 2.415 | | 14.8 | 6.08 | 12.105 | 6.006 | 5.528 | 0.043 | 33.917 |
| 20 February 2022 | **17.453** | 6.237 | 2.494 | | 31.3 | 22 | 7.35 | 7.741 | 7.521 | 0.045 | **123.917** |
| 31 March2022 | 11.382 | 5.848 | 2.424 | | 11.79 | **0.46** | **3.575** | **11.998** | **14.401** | 0.028 | 45.583 |
| 05 April 2022 | 11.466 | 6.604 | 2.826 | | 23.04 | 8.24 | 6.331 | 9.825 | 10.855 | 0.033 | 68.083 |
| 07 April 2022 | 10.761 | 6.769 | 3.121 | | 20.4 | 20.72 | 8.38 | 9.327 | 9.249 | 0.034 | 73.917 |
| 25 April 2022 | 8.522 | 8.224 | 1.938 | | 8.43 | 2.14 | 4.923 | 8.688 | 8.905 | 0.017 | 43.917 |
| 20. May 2022 | 5.774 | 10.361 | 2.379 | | 18.95 | 3.42 | 10.741 | 6.739 | 6.965 | 0.016 | 53.917 |
| 05 June 2022 | 4.548 | 10.457 | 2.229 | | 7.85 | 3.1 | 9.132 | 6.472 | 6.707 | 0.011 | 23.917 |

The DNT$_{30}$ index correlates negatively with discharge and DOC export, while the correlation between DOC concentration and the DNT$_{30}$ is positive. This seemingly contradictory behavior can be explained by considering the relative contribution of discharge and DOC concentration to the sum parameter DOC export. The relation between discharge and DOC concentration is rarely linear and leads to an overrepresentation of discharge in Eq. 11. Spectral slopes show significant



correlations (p < 0.1) with each other, with DOC export, and with the $SUVA_{254}$ index. The spectral slopes were additionally significantly correlated to event length and the $SL_{290}$ to EC. The internal correlation between DOC quality parameters can be explained by the nature of the indices (Hansen et al., 2016). While quantifying the strength of aromatic bonds ($SUVA_{254}$), the

molecular weight also decreases, when these bonds are broken, resulting in a weaker absorption at 254nm (Helms et al., 2008). Since the $SL_{275-295}$ and $SL_{290-350}$ are indicating the molecular weight (Helms et al., 2008), influenced by aromaticity, it is to be expected to see small slopes at high $SUVA_{254}$ values and vice versa. Table 2 shows this behavior where $SUVA_{254}$ highly negatively correlates with the spectral slopes $SL_{275-295}$ and $SL_{290-350}$. The strong negative correlation can therefore be explained by the near-zero value of the majority of the $SL_{275-295}$ and $SL_{290-350}$ slopes, which indicates that very slow

degradation is in progress (Hansen et al., 2016; Helms et al., 2013).

**Table 2 Spearman-Correlation-coefficient for mean values of observed parameters. Mean values were calculated per event. Significant combinations are highlighted by \*. Q = event discharge [l/s], T = mean water temperature [°C], DOC = DOC concentration [mg/l], EC = electric conductivity [µS/cm], P = aggregated precipitation [mm], API = API before event [-], $SUVA_{254}$ = specific ultraviolet absorption at 254nm [ L m$^{-1}$mg-C$^{-1}$], $SL_{275}$ = Spectral slope between 275 and 290nm [*10$^{-3}$nm$^{-1}$], $SL_{290}$ =**
**Spectral slope between 290 and 350 nm [*10$^{-3}$nm$^{-1}$], Cexp = Carbon export during event [kg], Length = event length [h]**

|        | Q      | T     | DOC    | EC    | P     | API   | SUVA254 | SL275  | SL290 | Lenght | Cexp |
|--------|--------|-------|--------|-------|-------|-------|---------|--------|-------|--------|------|
| Q      | 1      |       |        |       |       |       |         |        |       |        |      |
| T      | -0.28  | 1     |        |       |       |       |         |        |       |        |      |
| DOC    | -0.01  | 0.19  | 1      |       |       |       |         |        |       |        |      |
| EC     | -0.09  | -0.27 | *-0.62 | 1     |       |       |         |        |       |        |      |
| P      | *0.55  | 0.29  | *0.37  | -0.03 | 1     |       |         |        |       |        |      |
| API    | 0.06   | 0.14  | *0.36  | -0.13 | *0.52 | 1     |         |        |       |        |      |
| SUVA254| -0.27  | -0,23 | 0.12   | -0.09 | -0.28 | -0.14 | 1       |        |       |        |      |
| SL275  | 0.3    | 0.22  | -0.03  | 0.17  | *0.51 | 0.31  | *-0.81  | 1      |       |        |      |
| SL290  | *0.34  | 0.11  | -0.23  | 0.34* | *0.45 | 0.17  | *-0.77  | *0.93  | 1     |        |      |
| Lenght | 0.21   | *0.4  | *0.65  | -0.21 | *0.73 | *0.37 | 0.16    | 0.13   | 0.01  | 1      |      |
| Cexp   | *0.5   | -0.02 | 0.15   | 0.17  | *0.73 | *0.42 | *-0.35  | *0.53  | *0.51 | *0.47  | 1    |

The positive correlation of spectral slopes and DOC export indicates that the DOC during events with higher DOC-export is mainly composed of lower quality DOC. This can be attributed to a higher portion of water from the unsaturated zone during event discharge, which was also assumed for the catchment by Spill et al. (2023). DOC is degraded in the unsaturated zone

(Gabor et al., 2015), draining into the river during events with higher precipitation and discharge. The lack of significant correlations between $DNT_{30}$ or $AI_{60}$ with any of the DOC quality parameters stand in contrast to the study of Werner et.al. (2019), where all parameters were significantly correlated (Werner et.al., 2019). This discrepancy might be attributed to the mainly agricultural land use of the Nesselbach catchment or the focus on event-based analysis, leading to a lower number of observations. However, the lack of a significant correlation of DOC quality with $DNT_{30}$ and $AI_{60}$ indices indicates that there

must be a factor with lower frequency controlling the DOC quality signal, such as season. Figure 5 shows boxplots of the




pre-event, event and post-event mean values of DOC-concentration and quality. The events are sorted by season. The changes between pre-event baseflow and event discharge are strongest in the DOC-concentration during winter months and summer. Because there were no significant events in autumn, the season is not represented in Figure 5. While event discharge has a clear impact on the DOC-concentration during events in Winter and summer, there are less changes during

the spring months. DOC-concentration was expected to rise significantly during the summer period (Hinton et al., 1997; Laudon et al., 2004), but Fig. 6 shows that this increase is only marginal. This indicates that, despite higher DOC production in summer there is less activation of DOC-sources during summer events compared to events in the colder months. Since there is a generally higher DOC-concentration in the summer baseflow, the broad range of concentrations during winter events must be explained by the activation of DOC-sources by factors like higher pre-event wetness and the hydraulic

conductivity in the catchment. Increase of general DOC production controlled by rising temperatures throughout the year is not a suitable factor to describe the DOC-concentration dynamics for this catchment, which is uncommon (Winterdahl et al., 2016).

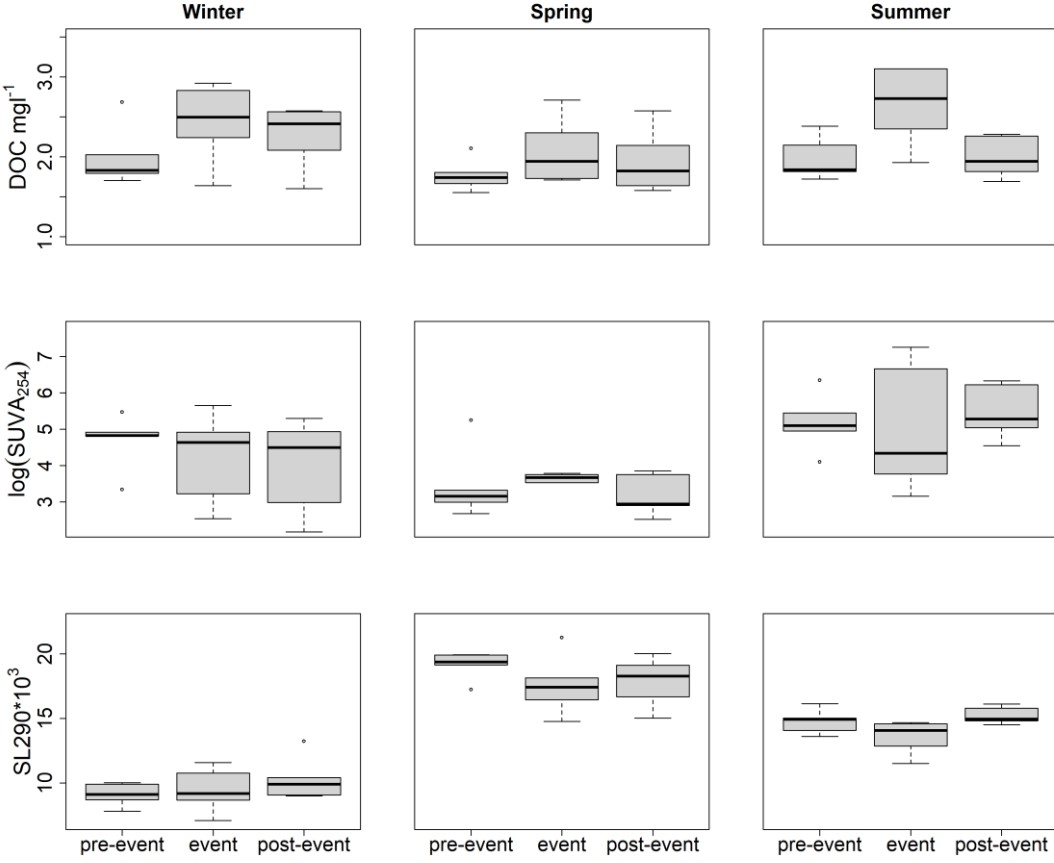

**Figure 5 Boxplots of pre-event, event, and post-event averages for the parameters DOC-concentration [mg/l] SUVA$_{254}$ (log-**
**transformed), and SL$_{290}$ multiplied by 1000 and separated by season. The SL$_{290}$ parameter also represents the SL$_{275}$ due to their strong correlation (R =0.95, p < 0.01).**



Changes in DOC-quality during events are represented in Fig. 5 by the log transformed $SUVA_{254}$ index and the $SL_{290}$. DOC-quality is affected by seasonal changes. Medians of $SUVA_{254}$ are lower during events in summer, but the boxplots show broader ranges than in winter and spring, especially during events. This fortifies the assumption, that additional sources of
DOC are activated during winter and summer and shows, that DOC-quality is more unstable in the summer months. Because DOC-quality tends to decrease with higher insolation (Jaffé et al., 2008; Christ and David, 1996) and higher degradation from greater biological activity driven by rising temperatures, it was expected that DOC quality changes are more pronounced during summer events. During the spring period, DOC-quality stays relatively stable and is less affected by events, expressing only a small range of DOC-quality in the boxplots.

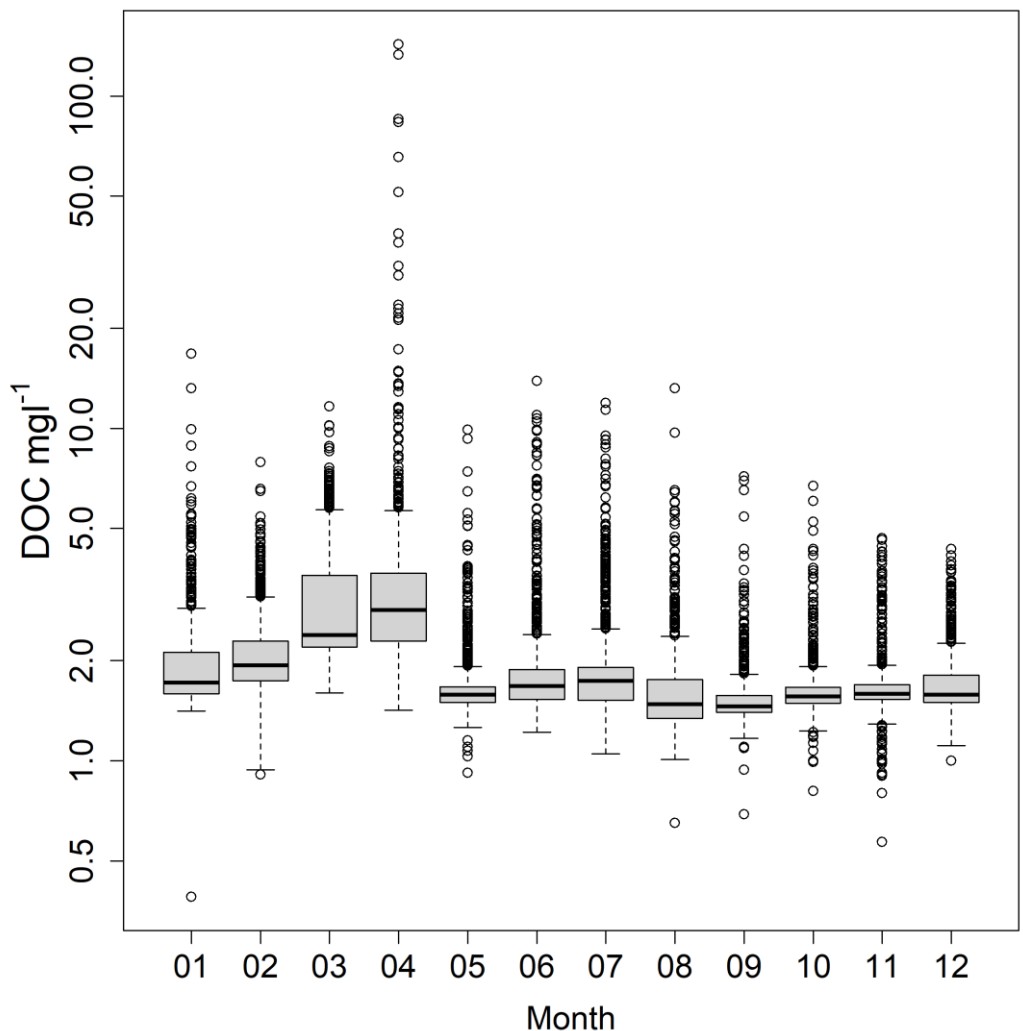


**Figure 6 Daily means of DOC-concentration aggregated and combined for monthly averages.**



One possible explanation for the observed high quality of DOC during baseflow in the summer months could be the production of fresh, and therefore higher quality, DOC in the riparian zone. This fresh portion of DOC gets mixed during the events with DOC from distal sources with a higher proportion of low-quality DOC, leading to the observed broad range of DOC-quality during events in summer. A greater wetness of the catchment during winter leads to the activation of DOC sources more distant from the stream, influencing the DOC-quality signal with a greater variety of DOC-species and therefore changing the frequency of the baseline signal during events (Werner et al., 2021). This results in a DOC quality mostly below the baseflow signal. On the contrary, there is less hydraulic activation of distant DOC sources during events in the summer months but a greater influence of degraded DOC flushing into the stream via overland flow. This observed behavior leaves the spring with the most stable DOC-quality signal. There are less additionally activated sources compared to winter and less biodegradation compared to the summer months. The greater range in DOC-quality due to the activation of different DOC sources during events can likely be attributed to the land use in the catchment (Stedmon et al., 2006; Graeber et al., 2012). Land use shifts the composition of DOC to an overweight of the DOC-component most likely produced in the corresponding land use. While the DOC-quality signal of the baseflow is likely dominated by riparian-produced fresh DOC in summer, the signal during events in summer and winter is more divers in its composition, resulting in a more unstable $SUVA_{254}$ signal. This explains the non-significant correlation of $SUVA_{254}$ to all climatological and hydrological parameters. These observations are further reinforced by the hysteresis analysis, as seen in Table 3. Clockwise hysteresis, which can be linked to near river sources of DOC (Vaughan et.al., 2017), are overrepresented during summer and late spring. In the colder months the number of anti-clockwise hysteresis is higher which can be interpreted as a higher proportion of distal DOC-sources (Vaughan et.al., 2017).

### 3.2 Carbon export

Figure 6 shows box plots of monthly DOC concentrations during the observation period. The monthly averages throughout the year vary between 1.58 mg l$^{-1}$ and 4.98 mg l$^{-1}$. In contrast, the median varies between 1.63 and 1.92 mg l$^{-1}$. The DOC-concentrations across the sampling period are not significantly changing from month to month when tested with a Wilcoxon-test for the location of the data. This aligns with the visual analysis of Fig. 5, that shows little change in median concentration during the monitored year.

Despite the clear logical connection between DOC-concentration and DOC-export, the correlation between the two parameters is small and lacks significance. This discrepancy is likely attributed to the practice of averaging event parameters to calculate event means. Although DOC-export is a sum parameter composed of the DOC-concentration and the discharge, the DOC-export dynamics follow a hysteresis loop, creating time lags and therefore suppressing an adequate evaluation of the spearman correlation test. Other parameters react more directly to the DOC-export. As Table 2 shows, DOC-export is significantly correlated to precipitation, event length, all quality parameters and $DNT_{30}$. Because these parameters are essentially sum parameters, they are more robust against time lags. Figure 7 shows a preference of higher cumulative DOC-load for clockwise events.



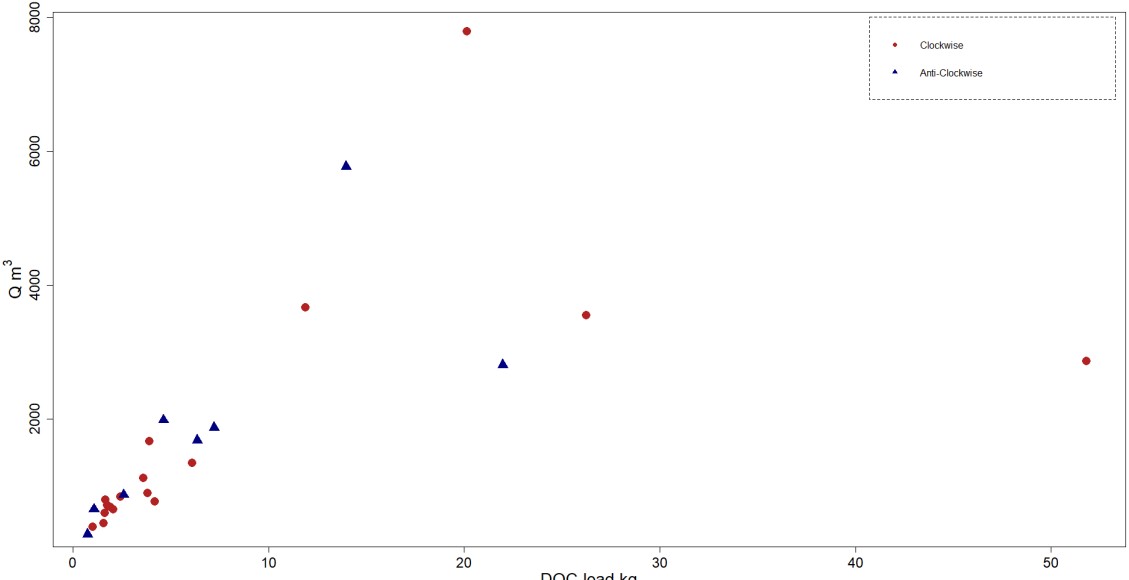

**Figure 7 Discharge and cumulative DOC-load during events.**

But this preference seems only small and there are not enough events to identify a significant clustering. The yearly export was estimated to 0.125 g (m²)⁻¹ a⁻¹ based on the available joint data of discharge and DOC-concentration. However, the total
carbon export of 0.125 g (m²)⁻¹ a⁻¹ is quite low compared to other studies (Alvarez-Cobelas et al., 2012), but the specific discharge (1.8 l s⁻¹ (m²)⁻¹) is similar. The precipitation events selected for this study, which represent 10% of the monitored time scale, accounted for 166 kg or 28% of the total carbon export during the entire measurement campaign.

This ratio between baseflow and stormflow runoff is quite typical for DOC-export when compared to all kinds of catchments (Bernal et al. 2002). The mean DOC-export is significantly lower when compared to a forested, equally sized catchment.
(Werner et al. 2019)) report for 0.17 g s⁻¹ of mean DOC-export while the mean export of the Nesselbach is by a factor of ten lower (0.0175 g s⁻¹). Table 3 shows the results of the HI analysis with an overweight for clockwise rotation, a minimum HI of -0.43 and a maximum HI of 1.179. The HI shows no significant correlation with the parameters $DNT_{30}$, $AI_{60}$, cumulative discharge, API or event length. This indicates that the direction of rotation and the magnitude of the hysteresis is mostly influenced by other factors. The distribution of rotation directions over the sampling period shows a high representation of
anti-clockwise hysteresis during the winter months and a clear overweight of clockwise rotation during spring and summer. Clock-wise rotation is commonly associated with the draining of near-river DOC-sources (Vaughan et al., 2017; Ducharme et al., 2021), and is expected to be more common in the warmer months, when DOC production is increasing. The more evenly distribution of rotation in the winter months indicates a more frequent activation of distal DOC sources like the forested part of the catchment and more distant slopes (Vaughan et al., 2017). Besides these patterns the distribution of the
HI over the sampled events is controlled by unknown factors. DOC export during events is higher in the warmer months (strongest in spring) than in the winter and autumn months.





**Table 3 Results of hysteresis analysis per event and the corresponding parameters total DOC export and total discharge. Values for the calculation of HI were not aggregated and hold an resolution of 5 min.HI = hysteresis Index [-], Rotation = direction of hysteresis (C = clock-wise rotation, AC = anti-clockwise rotation), C$_{EXP}$ = Carbon export during event [g], Q$_{cum}$ = cumulative event discharge [m³], Length = event length [h], API = antecedent precipitation index [-].**

| Date | Hi | Rotation | C$_{exp}$ | Q$_{Cum}$ |
|---|---|---|---|---|
| 27 January 2021 | 0.213 | C | 1553 | 443 |
| 08 February 2021 | -0.048 | AC | 1068 | 655 |
| 09 March 2021 | 0.29 | C | 3913 | 1668 |
| 11 March 2021 | -0.117 | AC | 13957 | 5768 |
| 09 APril 2021 | 0.081 | C | 11882 | 3663 |
| 10 May 2021 | 0.589 | C | 1915 | 687 |
| 21 June 2021 | 0.0195 | C | 4184 | 766 |
| 24 June 2021 | 0.16 | C | 2062 | 657 |
| 28 June 2021 | 0.227 | C | 3823 | 892 |
| 01 July 2021 | -0.024 | AC | 2590 | 871 |
| 08 July 2021 | 0.337 | C | 2433 | 842 |
| 13 July 2021 | 0.404 | C | 26244 | 3546 |
| 03 August 2021 | 0.646 | C | 1657 | 798 |
| 12 November 2021 | -0.302 | AC | 742 | 282 |
| 02 January 2022 | 1.1612 | C | 1621 | 603 |
| 01 February 2022 | -0.599 | AC | 6350 | 1687 |
| 06 February 2022 | 0.026 | C | 1757 | 718 |
| 16 Feburary 2022 | -0.695 | AC | 4622 | 1986 |
| 19 Feburary 2022 | 0.215 | C | 20125 | 7785 |
| 31 March 2022 | -0.049 | AC | 7216 | 1871 |
| 03 April 2022 | -0.038 | AC | 21965 | 2811 |
| 07 April 2022 | 1.179 | C | 51794 | 2867 |
| 25 April 2022 | 0.432 | C | 6107 | 1349 |
| 20 May 2022 | 1.736 | C | 3582 | 1121 |
| 05 June 2022 | 0.428 | C | 999 | 392 |
|  |  |  |  |  |
| Mean |  |  | 8166 | 1789 |
| Max | 1.179 |  | 51794 | 7785 |
| Min | -0.43 |  | 742 | 282 |



The average temporal pattern of DOC-concentration, as shown in Fig. 6, is unexpected. Typically, warmer temperatures are associated with higher overall DOC production (Jawitz and Mitchell, 2011; Jiang et al., 2014). In this study, the medians of DOC concentrations in winter slightly exceed those of the summer months. While events in summer show slightly higher DOC-concentrations on average during events than events in winter (see Fig. 5), the overall export in summer is lower due to lower concentrations in baseflow.

The relatively low DOC-concentration in summer and the resulting low carbon export can be attributed to the land use characteristics of the catchment. While many studies focused on forested catchments, where the carbon export is significantly higher (Werner et al., 2019), catchments dominated by agriculture are less frequently monitored and tend to exhibit lower DOC-export (Li et al., 2015; Worrall et al., 2012). Since plants are harvested from farmlands, the carbon fixated in those plants is not available for export via the catchment outlet. Additional factors explaining the lack of carbon in this catchment might be a reduced half-life of DOC due to mineralization by photodegradation and a transformation in the groundwater (Pabich et al. 2001), (Osburn et al., 2009; Hossler and Bauer, 2012). It's also worth considering the impact of our high-resolution, in-situ sampling method, as it provides a more accurate representation of DOC-export patterns and loads than low-frequency automatic sampling or grab samples (Etheridge et al., 2014; Werner et al., 2019).

The main extrinsic driver for the increase in DOC export in this study is precipitation (Table 2). Because events with a 2-fold rise of discharge from pre-event baseflow are rare, the impact of average precipitation events on the overall C-export in the catchment is not very high, leaving the maximum values of DOC-export to a few selected events with at least 3-fold higher event water than pre-event baseflow discharge values, as show in Tab. 1. Besides the anticipated crucial role of discharge and precipitation, correlation analysis shows that also event length, $DNT_{30}$ and API play a significant role for the DOC-export. This might be due to higher DOC-mobilization in the soils during longer, but not necessarily more intense precipitation events. Short and high-intensity precipitation events do not mobilize as much DOC as longer, but less intense events. API is a measure of soil moisture and another driver of DOC-export, with a positive correlation of 0.36. The API explains a fair portion of the DOC-exports during events. Thus, events with a higher pre-event wetness have significant higher mean DOC-export (Tunaley et al., 2017). $DNT_{30}$ correlates negatively with DOC export meaning that at a greater temperature to discharge relation, has a negative impact on the DOC export due to smaller discharge values.

## 4 Conclusion

Measuring DOC and discharge in 5 min timesteps provided valuable data for applying correlation-analysis and hysteresis-analysis, making it easier to define the starting point and evaluating the form of the hysteresis. This resulted in detailed insights into the DOC-export dynamics and the changing patterns of DOC-quality during precipitation events.

Besides the internal correlation between the DOC-quality parameters, the changes in the quality signal are mostly driven by DOC-export, precipitation, and discharge. Since DOC-export is a proxy for higher precipitation and DOC-concentration, the changes in the DOC-quality over all events can mostly be attributed to seasonal changes and land use characteristics. The

most important contributing source for the lower quality DOC is most likely the surface runoff from tillage and farmland.

The mostly clockwise rotation of the discharge-export hysteresis, which indicate near river DOC sources, amplify this assumption. Although its source is in a mixed forest, the greater part of the Nesselbach is mostly surrounded by agriculture adjacent to the riverbed, serving as fast reacting DOC source. The surface runoff from the farmland might shift the DOC composition of the stream water towards lower $SUVA_{254}$ values.

The findings suggest that the DOC-export from the Nesselbach-catchment is low on a yearly average but can still reach high

loads during heavy precipitation events. The variation of DOC concentration throughout the year follows an unexpected pattern, with nearly stable medians per month but high mean concentrations during precipitation events in winter and summer. Whether this phenomenon is attributed to the harvesting of crops or the refreshing of yet unknown DOC-sources in the unsaturated zone is yet to be investigated.

**Code and data availability**

Field data and processing algorithms are available upon request from the author.

**Author contributions**

MG conceptualized the research. CS and LD collected the data. LD analysed the data with contribution from CS. LD wrote the manuscript. All coauthors reviewed and edited the manuscript.

**Competing interests**

The authors declare no competing interests.

**Acknowledgements**

The authors acknowledge the support provided by the laboratory of Siedlungswasserwirtschaft at the University of Kassel, as well as the assistance of their research assistants during data collection and laboratory analyses.

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
