# Peer review of "Event based high resolution measurement of DOCconcentration and quality in a rural headwater catchment."

_EGUsphere, 2024_

## Author Comment (AC2)

**Introduction**

*The introduction is missing key literature support and justification for this study. Why is there a need for long term measurements? How can this help improve carbon export from these small catchments? Why are agricultural systems important for DOC composition and fluxes? What information is already available for the influence of agriculture on DOC composition? There is plenty of literature on this topic already. Seems the discussion is focused on big rivers, but the study is in a small catchment. What is the importance of small agricultural catchments, and how might their characteristics/hydrology impact DOC fluxes? What is hysteresis and why is it important? How can it be used to answer DOC flux questions?*

- Additional literature will be provided and the significance of the dataset will be more highlighted. The explanation for hysteresis will be elaborated in the method section.
- Literature for hysteresis and broader explanations in the introduction will be provided.

*The manuscript defines DOC as "good" or "bad" quality – "good" quality being highly aromatic/resilient DOC and low quality being DOC that is broken down and is more prone to other processes. I strongly challenge this notion that there is a "good" vs "bad" quality DOC and the use of quality here is strongly misleading. For instance, DOC is introduced as "good" quality, but in the same sentence says that same "good" quality is bad for water quality. So in that vein, what does quality actually mean? It is important to note that DOC behaves differently depending on the biogeochemical process. I might make the argument that the same highly aromatic "good" quality DOC defined in this paper is actually "bad" quality for microbial processes since these compounds are typically not what microbes like to eat. So, it is my recommendation to remove any reference to "quality" as it is to subjective and confusing. I recommend using the term "composition" instead*

- Based on the literature there is an ambivalent behavior of DOC, there is high quality DOC which can lead to challenges for WWTP and there is bad quality DOC which can lead to different challenges for WWTP. The recommendation to change quality to composition is highly welcomed to minimize any confusion.

*Line 83 (Major aims): What is meant by a "relevant precipitation event"?*

- Relevant precipitation events mean any event over a threshold of 7 mm precipitation. These events were strong enough to significantly increase the discharge compared to baseflow. This threshold is not arbitrarily chosen but is rather based on the experience gained during our field trips. Events below 7 mm did not significantly increase discharge during our monitoring period.

**Methods**

*Study Site: It is not defined here that this is in Germany. Germany is only mentioned in abstract. Figure 1 needs more geographical context for non-German/European readers. Please add a map of Germany with the location of the catchment within and highlight any relevant features needed for context – for instance the broader watershed or the "European uplands" as mentioned in the manuscript.*

- It will be no problem to add additional information and orientation.

*From what time frame does the study represent?*

- Timeseries starts in February 2021 and ends in July 2022, this information will be added.

*In situ measurements: The manuscript uses data collected from the spectro::lyzer (S::CAN) and uses the "universal calibration" set forth within the instrument to obtain DOC fluxes. There are a number of inherent challenges with this:*

- We will provide the biweekly grab samples and available event samples we used for calibration.
- Turbidity in the catchment depended on the season and the intensity of precipitation events, some events were strongly dominated by overland flow and had very low turbidity. We will reevaluate the data with turbidity correction. However, with a validated calibration using lab-analysed grab samples, we are not sure if this will significantly improve the accuracy of our time series.

*DOM optics: Sr and SUVA are properties of chromophoric DOM (CDOM). They are not representative of the entire DOC pool. So, Sr for instance cannot be stated as an indicator of DOC molecular weight because there are non-light absorbing components of the DOC pool. It is an indicator of CDOM molecular weight. Furthermore, Sr is not exclusively an indicator of photochemistry. Yes, photochemistry can impact Sr, but Sr is also impacted by source and other biogeochemical processes. This should be clarified.*

- Clarification will be provided.

*Discharge and Smoothing: It was unclear to me how this Nivus ultra sonic probe was used to calculate discharge for open streams? Some additional information on the setup, how the measurements were collected, and discharge extrapolated is needed. A conceptual figure could be beneficial. There were also a number of steps taken for discharge smoothing. I think a Supporting Information with some additional discussion and figures would help with this as it was hard to follow based on the text alone.*

- All of the equipment, including the nivus probe was located at the end of a concrete tube with a known diameter. This was necessary to operate the Nivus probe since it relies on the reflected ultra sonic waves to function correctly. The description of the monitoring setup will be clarified. We will also provide an example plot of discharge smoothing, or a flowchart of the algorithm for the supplement.

[Figure]

*Figure 1 Example picture of flowchart for discharge smoothing algorithm.*

*An explanation of metrics by which storms were delineated and chosen to include in statistical analyses is needed.*

- Explanation of storm delineation will be provided.

*Need a section explaining statistics used*

- We will provide a section for used statistics.

**Results and discussion**

*At a high level, I recommend putting DOC concentration first then composition. Apart from the challenges already listed above, the major challenge with this section is that there is no clear novelty. The primary finding is that DOC loads increase with precipitation. There are further inferences that land cover may be important, however, this study was not designed to test the impact of agriculture directly and comparative references within the literature that could justify this conclusion are minimal. The generalized interpretation of hysteresis indies are valid, but it is unclear there broader utility for answering key questions related to this dataset.*

- Recommendation to put DOC-concentration at first is very valid ad will be considered. Literature to compare land use will be increased.
- Research questions will be more refined.

**Minor comments**

*There are many references to significant correlations throughout without any statistics associated with the statements.*

- Details on significance will be provided, including used test (spearman correlation)

*line 245: Unclear why this is considered an outlier based on text.*

- Mean discharges per event will be provided.

*line 255-258: Novelty?*

- We will provide additional literature.

*Line 270: overrepresentation of discharge in calculation of flux? Unclear what is meant by this?*

- In most cases the increase of discharge is way higher than the increase in DOC-concentration, sentence will be revised for clarification.

*line 273-275: wording, unclear the interpretation.*

- Sentence will be revised.

*line 278-280: wording, explanation again unclear.*

- Sentence will be revised.

*Line 290: The parameters DNT30 and AI60 need further explanation on why they are linked with DOC in other studies and what that means. It is unclear how one would interpret a relationship between these indices and DOC parameters.*

- Additional explanation will be provided.

*line 300: Is there an explanation for why DOC might be different for the spring?*

- We will check if there are hydrological explanation like discharge volume or differences in precipitation intensity.

*line 300-308: It would be more appropriate to incorporate more multi-way statistical approaches when discussing multiple different controls on DOC.*

- We will double check if for example ANOVA will provide more sophisticated results.

*line 315: Higher insulation in the summer?*

- Spelling error: We will change it to irradiation.

*line 330: discussion of riparian vs distal sources… are there any other measurements in the catchment that would support this? Such as soil data?*

- We will double check if there are organic carbon values available from government or other sources.

*line 334: Unclear wording – and unclear based on the study design how land cover could be stated as important.*

- Sentence will be revised, and literature added.

*Line 338-341: I feel like there is something here that could be better explored in disentangling the seasonal differences. I encourage authors to expand on this more with more complete interpretations.*

- We will check potential seasonal changes in catchment connectivity.

*Line 387: Again, unclear how land use can be attributed as a contributor of DOC here as this was not a part of the study design, and the manuscript comparisons are with only 1 other forested study. This is not enough to make conclusive statements.*

- A broader review of literature will be conducted.

**Figures**

*Figure 2a. I recommend adding directionality to this figure with some arrows so we can more clearly see the trend.*

- Will be done.

*Figure 2b is labeled as DOC but the legend says nitrate. Note that if Figure 2b is in fact DOC, this would be anticlockwise hysteresis, contrary to what is stated in Figure 2a. Please correct and clarify.*

- Will be corrected.

*Figure 3 and Figure 6 would benefit from being a single annotated figure.*

- To be considered

*Figure 4 is illegible in the current pre-print format. I might also recommend finding ways to delineate the storms of interest on this figure if possible.*

- Storms of interest will be delineated.
- Figure will be reformatted for better legibility.

*Figure 5: It is unclear what this figure represents. Is this an average of all storms during the designated seasons or is it one storm? How are the storms delineated as pre- and post- event? Is there a consistent metric for changes in discharge that defines the beginning and end of an event?*

- We will expand the description, threshold for storm is the activation and deactivation of the automated sampler. This will be explained in more detail in the method section.

*Figure 7: It is unclear what is meant to be represented by this figure. I might recommend an alternative figure, one that uses a DOC hysteresis index vs a flushing index, which could then be color coded by DOC load or season. I think this would provide a better understanding of the hydrological conditions and controls on DOC export of the seasons. There is an example of this in the Vaughan et al. reference above.*

- Thank you for the helpful advice, we will take the suggested alternative figure in consideration.

*Table 1-3: Unclear what these tables represent. Is this for the entire event or just at the peak? Better explanation of storm delineation is needed as suggested in the comment above. I might also recommend the authors consider more creative ways to display this data for the main text and move these tables to a supporting information.*

- Tables 1-2 represent the mean value of a given parameter over a whole event.
- Values and parameters of table could alternatively be presented as boxplot and highlighting of significant values in the written text, or by a panel of plots where every parameter is plotted against events.

- Table 3 could alternatively be plotted as timeseries with colored dots, indicating the rotation of hysteresis, or visualized by clustering the rotation of hysteresis by season.